# Robotic versus Laparoscopic Partial Nephrectomy in the New Era: Systematic Review

**DOI:** 10.3390/cancers15061793

**Published:** 2023-03-16

**Authors:** Estefanía Ruiz Guerrero, Ana Victoria Ojeda Claro, María José Ledo Cepero, Manuel Soto Delgado, José Luis Álvarez-Ossorio Fernández

**Affiliations:** Urology Department, Hospital Universitario Puerta del Mar, 11009 Cadiz, Spain

**Keywords:** partial nephrectomy, robotic, laparoscopic

## Abstract

**Simple Summary:**

The aim of the review is to investigate the differences between RAPN and LPN, both in terms of functional and oncologic outcomes, in the new era of minimally invasive surgery.

**Abstract:**

(1) Background: In recent years there have been advances in imaging techniques, in addition to progress in the surgery of renal tumors directed towards minimally invasive techniques. Thus, nephron-sparing surgery has become the gold standard for the treatment of T1 renal masses. The aim of this study is to investigate the benefits of robotic partial nephrectomy in comparison with laparoscopic nephrectomy. (2) Methods: We performed a systematic review according to the PRISMA criteria during September 2022. We included clinical trials, and cohort and case-control studies published between 2000 and 2022. This comprised studies performed in adult patients with T1 renal cancer and studies comparing robotic with open and laparoscopic partial nephrectomy. A risk of bias assessment was performed according to the Newcastle—Ottawa scale. (3) Results: We observed lower hot ischemia times in the robotic surgery groups, although at the cost of an increase in total operative time, without appreciating the differences in terms of serious surgical complications (Clavien III–V). (4) Conclusions: Robotic partial nephrectomy is a safe procedure, with a shorter learning curve than laparoscopic surgery and with all the benefits of minimally invasive surgery.

## 1. Introduction

In recent decades there has been an increase in the incidence of renal cancer, due to the more frequent use of imaging techniques [1], leading to progress in the development of minimally invasive surgical techniques, which has positioned nephron-sparing surgery as the gold standard for the treatment of T1 tumors, bilateral renal masses or renal neoplasms in single-kidney patients [2], to try to preserve renal function, compared to radical nephrectomy; however, partial nephrectomy (PN) is a challenging technique that requires an adequate preoperative study [3].

In recent years, the development of minimally invasive surgery has made robotic-assisted partial nephrectomy a safe technique that has reduced the warm ischemia time (WIT), compared to the laparoscopic approach [4]. The robotic approach is used in many other urological surgeries such as prostatectomy and it is widely developed in many other fields; for example, breast cancer and reconstruction surgery [5,6].

Since the first RAPN was performed by Gettman et al. in 2002 [7], this technique has evolved to be able to treat patients with T2 tumors or complex masses, due to the development of vascular reconstruction using 3D technology [8] for preoperative planning and surgical simulation [9].

The aim of the study is to investigate the differences between RAPN and LPN, both in terms of functional and oncologic outcomes.

## 2. Materials and Methods

This systematic review has been prepared according to the Preferred Reporting Items for Systematic Reviews and Meta-Analysis (PRISMA) criteria [8]. It was performed in September 2022, using the search terms: “robot” OR “robotic” AND “partial nephrectomy” AND “robotic nephrectomy advances” in Pubmed and the Cochrane Library. This systematic review was registered in Research registry with this identifying number: reviewregistry1565. Ethics approval was not required for this work since it does not involve human subjects. Two independent authors independently assessed each included study and differences or debates were resolved by consensus.

### 2.1. Inclusion and Exclusion Criteria

Studies with the following criteria were included: randomized clinical trials; cohort studies and case-control studies, published between 2000 and 2022; studies conducted in adult patients with T1 renal cancer; and studies comparing RAPN with open partial nephrectomy (OPN) or with LPN (Figure 1).

We excluded studies on radical nephrectomy, or comparing laparoscopic and open partial nephrectomy; books or manuals; editorials; letters to the editor; comments; clinical cases; unpublished articles; and conference abstracts, as well as studies without available data.

### 2.2. Data Mining

Data were extracted independently by 2 authors; differences were negotiated by both authors.

### 2.3. Risk of Bias Assessment

The risk of bias assessment was performed according to the Newcastle—Ottawa Scale (NOS) for nonrandomized trials. Two independent reviewers assessed the risk of bias of all included studies according to the NOS. Any inconsistencies were discussed and resolved by agreement. A total score of 5 or less was considered low, 6 to 7 was intermediate, and 8 to 9 was high quality.

### 2.4. Data Analysis

Data were extracted using a predefined data extraction form. Baseline demographic data (age, tumor size, baseline renal function, and baseline CKD), perioperative data (operative time, estimated blood loss, complications, surgical margins, and hospital stay) and functional (estimated glomerular filtration rate [eGFR] decline) and oncologic data (cancer-specific survival, recurrence free survival, and overall survival) were extracted from the studies when available.

### 2.5. Data Synthesis

Given the great clinical and methodological heterogeneity in the patient population and procedures, we qualitatively synthesized the results and developed a descriptive summary for each procedure.

## 3. Results

### 3.1. Selection of Studies

In the review, 1079 studies were found. After eliminating duplicate studies, a total of 1067 studies were evaluated, of which 1030 were eliminated after reviewing the titles and abstracts, leaving 37 studies to be evaluated. Out of the 37 studies, 17 met the inclusion criteria and 9 were excluded based on text quality criteria and ineligible outcomes. Finally, we selected 8 studies that met the inclusion criteria (Figure 1).

A total of 2705 cases were included in the review (558 robotic, 522 laparoscopic and 1625 open). Four studies compared the results between open, laparoscopic, and robotic partial nephrectomy, while 2 of them compared the laparoscopic vs the robotic approach; one of them evaluated only the laparoscopic approach and the last one assessed the robotic approach.

Most of the studies reported on the experience of a single center, except for three of them [10,11,12] and the most frequent design was the retrospective study (two of them with prospective data collection [10,11], except for two cases which were prospective studies and one of them a randomized controlled trial [12,13], being the main patient recruitment periods between 2001 and 2020 (Table 1).

PRA: PROSPECTIVE RANDOMIZED.; R: RETROSPECTIVE.; RP: RETROSPECTIVE BUT PROSPECTIVE COLLECTION. Matching: Age (1), Sex (2), Body Mass Index (3), ECOG (4), side of lesion (5), location (6), characteristics (7), nephrometric scales score (8), preoperative creatinine (9), preoperative hemoglobin (10), type of approach (11), ischemia time (12), estimated blood loss (13), surgical time (14), intraoperative and postoperative complications (15), length of hospital stay (16), positive margins (17), occurrence of renal failure (18), follow-up period (19), cancer-specific survival (20), recurrence-free survival (21), overall survival (22)

### 3.2. Analyzed Data

Demographic characteristics, preoperative intraoperative and postoperative parameters included in the 8 studies are described in Table 2 and Table 3. A higher BMI can be observed in the patients included in the robotic surgery group in most of the cases. The treated tumors had a maximum stage T1b (smaller than 7 cm), being mostly endophytic and also renal hilium tumors. Most studies report data on complications, surgical time, warm ischemia, hospital stay, or changes in glomerular filtration rate. However, only one study provides data on cancer specific survival and metastasis free survival, what could conduce a risk bias in the interpretation of the final data. On the other hand, no study detailed the access route (retroperitoneal or transperitoneal) used to perform the surgery, which could conduce a bias because of the location of the tumors, being a relevant factor to choose a different approach. Referring to the histological specimen, a malignant histology will be the most frequent, with clear cell renal carcinoma being the most constant in the work of Haseebuddin, Zaid, Ingels, and Würnschimmel. Haseebuddin et al. are the only ones to separate patients with tumors smaller than 2 cm from those with tumors larger than 2 cm.

In relation to the warm ischemia time, we observed that in robotic surgery the range oscillates between 18 and 24.7 min, the latter figure being more common with tumors that are more complex to resect and larger than 2 cm. On the other hand, the interval in laparoscopic surgery ranges from 21 to 24 min. With regard to the total surgical time, it was superior in robotic surgery compared to laparoscopic and open partial nephrectomy.

## 4. Discussion

PN offers a better preservation of renal function than RN, which reduces the incidence of metabolic and cardiovascular disorders, making PN a better option for the treatment of small tumors. The main objective of PN is to achieve favorable functional results while maintaining oncologic safety, equivalent to radical nephrectomy in small renal masses (cT1). However, nowadays there are a growing number of publications that even support nephron-sparing surgery for larger tumors whenever technically feasible (cT2), without finding significant intraoperative or major postoperative complication differences with radical nephrectomy, nor in the overall survival at 5 years or in the disease-free survival rate [18]. In this way, RAPN adoption has increased in the last decades, and indications have been expanded also to more complex and challenging cases [19,20].

Although PN can be safely performed through multiple approaches in our population, regardless of BMI and age group, and is also a safe procedure in elderly patients, each technique has its own set of advantages [10].

In the present work, our aim was to evaluate the role of RAPN in the treatment of renal tumors smaller than 7 cm, and to objectively compare RAPN with LPN and OPN. According to our review, no differences were observed in the different peri- and postoperative variables analyzed between the different minimally invasive techniques (RAPN and LPN). Although, we observed a greater tendency to a lower WIT, a lower blood loss and a lower postoperative EGFR difference with RAPN. However, we have detected a higher complication rate with OPN, as well as higher blood loss and a higher EGFR difference, compared to RAPN. However, only two of the papers are based on randomized studies, and there is high variability between papers.

According to Haseebuddin, WIT is inferior in RAPN versus LPN in the surgery of small renal masses, with a more significant advantage in the treatment of hilar tumors. In contrast, longer WITs were observed in the treatment of tumors larger than 2 cm [17] It seems that RAPN favored the reduction of WIT, with the widespread diffusion of a minimally ischemic approach [21,22]. In this context, the standardization of the description of the anatomic features of renal tumors has resulted in the development of multiple scoring systems [23]. Despite the fact that the vast majority of studies included in this review have lacked a standardized surgical complexity tumor score, it has been shown that the tumor location can increase surgical difficulties, and this could affect perioperative variables [21]. However, only one of the papers presented a nephrometry score. Hayn et al. [23] reported that a higher nephrometry score is associated with increased blood loss, higher WIT and a longer hospital stay, as assessed in a group of patients undergoing LPN.

The ischemia time has been considered throughout history as a fundamental determinant in postoperative WIT; a WIT of <25 min being the widely recommended standard at which any acute kidney injury is considered reversible. Multiple studies have shown worsening functional outcomes associated with WIT > 25 min [23,24]. Therefore, the robotic approach probably favors the postoperative preservation of EGFR, improving warm ischemia times compared to laparoscopy [12]. Currently, some brand-new techniques that are known as “zero ischemia time” procedures have been described. Their main feature is to minimize renal damage, achieving a shorter WIT. This performance is also widely used in robotic surgery such as a clampless surgery or selective and supraselective clamping [25], and its development is related to the use of near-infrared fluorescence (NIRF) with the FireFly system using indocyanine green (ICG) or with the help of intraoperative ultrasound; both strategies supply renal pedicle and nutritional-vessels tumor identification [26]. NIRF with ICG is considered a safe and practical tool that helps the surgeon to identify anatomical structures and resection margins, enhancing the renal function’s short-term preservation [27,28]. Nonetheless, in this systematic review, only a few studies describe these tools and there is no consensus on their use. According to Yang et al., preoperative renal function, preserved parenchymal volume, and WIT affect short-term postoperative renal function outcomes. Their study demonstrates that as a result of using RAPN with the aid of ICG, the WIT could be minimized, preserving a more healthy kidney parenchyma, and therefore achieving exceptional renal function preservation and a lower incidence of postoperative complications [28]. In this background, when a significant decrease in renal function is detected after PN, it is not only due to ischemia-reperfusion damage after clamping the renal hilum, but also due to the resected healthy parenchymal tissue or a reconstructive lesion from renorrhaphy [29].

Another important factor to highlight is the presence of positive surgical margins in PN, which can worsen survival. According to Porpiglia, RAPN and LPN showed a lower presence of positive margins versus OPN [14], being higher with RAPN versus LPN according to some of the studies analyzed [13,14]. Nowadays, the trendiest surgical resection in PN is tumor enucleation, which has been favored according to different meta-analyses with a lower rate of complications, blood loss, transfusion needs, and a shorter hospital stay. On the contrary, this surgical fashion could increase the positive margins rate, but there is no evidence of a higher rate of tumor recurrence in this way [30].

Regarding the presence of postoperative complications, no differences were found between RAPN and LPN corresponding to this systematic review, and according to Ingels et al. [16] the rate of severe complications (Clavien Dindo III-V) in the RAPN group is about 6.2%.

Hung introduced the term Trifecta, with the aim of defining an ideal surgical outcome in PN. Trifecta is considered a new term to describe the surgical quality that combines the following characteristics: the presence of negative surgical margins, the absence of serious complications, and a ≤30% postoperative estimated glomerular filtration rate (eGFR) reduction [28]. Trifecta has been widely employed in the literature as a key indicator of PN success; as a consequence, different scoring systems have been proposed including anatomical complexity tumor classifications, which could predict perioperative outcomes [31].

If we compare both of the minimally invasive approaches, we find that the robotic approach provides advantages over laparoscopic surgery, such as a surgical field three-dimensional vision, as well as allowing the surgeon to emulate open surgery movements due to a greater range of motion and tremor elimination. Robotics also provide greater ergonomics and greater comfort to the surgeon regarding the performance of the procedure. However, there are also disadvantages, such as haptic sensation loss or higher procedures costs. From this point of view, the Da Vinci robot service is expensive, with annual maintenance costs approaching 100,000 euros [32].

Today, technology applied to robotics is gaining a relevant role in the development of 3D models and augmented reality images or holograms that are used in preoperative planning, contributing to better control of anatomical and vascular tumor relationships. The recent evolution in imaging techniques, with the state-of-the-art ultra-high resolution CT (UHR-CT) development, supply accurate anatomical images that help to identify urinary tract invasion, vascular structures, and the renal pedicle, which are obtained during early or late arterial and excretory phases, being transferred to a workstation that processes information to obtain 3D models. These UHR-CTs improve conventional CT spatial resolution [normal resolution (0.5 mm × 80 rows/896 channels), high resolution (HR; 0.5 mm × 80 rows/1792 channels), and ultra-high resolution (0.25 mm × 160 rows/1792 channels)30), helping with 3D model preoperative planning which shows a lower collecting system injury rate, a shorter surgical time, and less ischemia time [32]. The most recent advance in robotics is real-time navigation with the TilePro system, which consists of superimposing 3D reconstruction images on the surgical field, serving as a guide for dissection and vascular and tumor identification [33].

Concerning the learning curve of these minimally invasive procedures, it seems to be longer in the laparoscopic approach [34]. However in the different studies that were included in this systematic review, there is a great variability in the surgeon’s experiences, so we have not been able to take into account the learning curve in our work. Although, Haseebuddin et al. reported that the learning curve to reduce WIT is 26 cases, and to reduce the total surgical time is 16 cases [17].

As an alternative to small-sized renal tumor surgery, different minimally invasive strategies such as focal therapy have emerged, which are mainly used in elderly patients or high-risk comorbid patients who are unfit for surgical treatment. One of the most popular is percutaneous cryoablation (PCA) [35], which is surrounded by controversies in the literature. While some studies suggest that RAPN and PCA have similar oncological outcomes in patients with small renal masses [36], others report that the recurrence rate is significantly higher after PCA vs RAPN [37], so it is essential to individualize the technique according to each patient and their neoplasm.

Most of the existing evidence on RAPN comes from non-randomized retrospective studies, which is one of the limitations of the present work. This systematic review includes only two prospective studies [12,13], one of them is randomized [13]. Moreover, only one of the studies mentions its inclusion criteria [11], and most of them have a lack of long-term follow-up, with only five studies that analyze the presence of positive margins. Furthermore, only one of the included studies reports cancer-specific and recurrence-free survival data [14]. As a result, a publication bias based on the current studies is relevant, as are variations in inclusion criteria and treatment protocols or use of different surgical techniques and assessment.

## 5. Conclusions

Robotics is an outstanding alternative to laparoscopy to perform a partial nephrectomy, which seems to provide great benefits for the treatment of complex tumors, leading to a WIT decline. RAPN also seems to improve oncologic control, surgeon ergonomics, and freedom of movement without increasing complications or worsening postoperative recovery. Finally, essential to the procedure planning, according to the surgeon’s experience, are patient characteristics and tumor morphology.

## Figures and Tables

**Figure 1 cancers-15-01793-f001:**
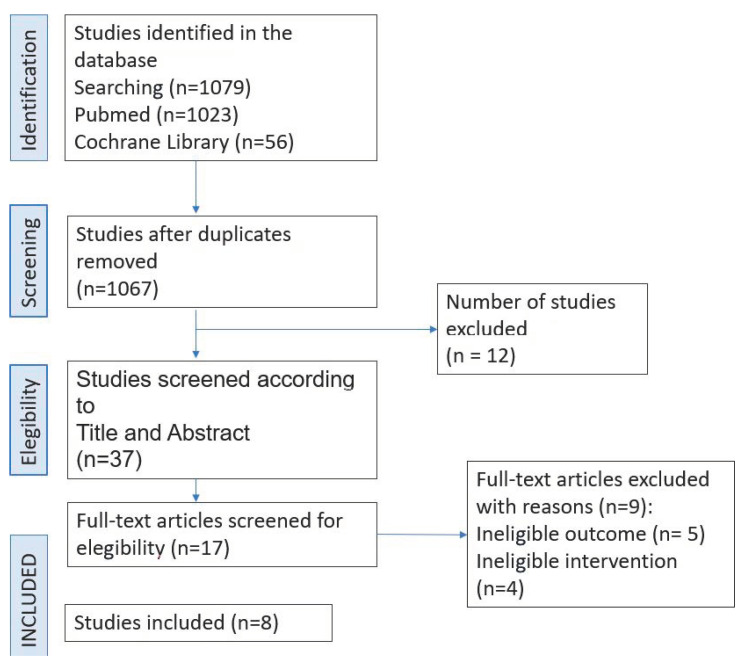
Study selection process.

**Table 1 cancers-15-01793-t001:** Characteristics of the included studies and quality assessment.

Studies	Level of Evidence	Design	No. of Centers	Recruitment Period	No. Surgeons	Coincidence	Follow-Up (Months)	Quality SCORE (NOS)
Porpiglia et al., 2015 [10]	3	RP	19	2009–2012	Multiple	1, 2, 3, 4, 5, 6, 7, 10, 11, 12, 13, 14, 15, 16, 17, 19, 20, 21	3	8
Webb et al., 2015 [14]	3	R	1	2005–2011	Multiple	1, 2, 5, 6, 7, 10, 11, 12	Perioperative	7
Kizilay et al., 2019 [15]	3	R	1	2012–2018	N/A	1, 2, 3, 5, 6, 7, 9, 11, 12, 13, 14, 15, 16, 17, 18, 19, 21, 22	60	8
Zaid et al., 2017 [16]	3	R	1	2001–2012	N/A	1, 2, 3, 4, 5, 6, 7, 17, 19	1	6
Ingels et al., 2021 [11]	3	RP	15	2000–2016	Multiple	1, 2, 3, 4, 5, 6, 7, 8, 9.10, 11, 12, 13, 14, 15, 16, 19	3	8
Würnschimmel et al., 2020 [13]	2	PRA	1	2015–2019	1	1, 2, 3, 5, 6, 7, 8, 9, 10, 11, 12, 13, 14, 15, 16, 17, 18	6	8
Haseebuddin et al., 2010 [17]	3	R	1	2007–2008	1	1, 7, 12, 13, 14, 16, 18	Perioperative	5
Hinata et al., 2020 [12]	2	PRA	22	N/A	Multiple	1, 2, 3, 6, 7, 9, 11, 12, 13, 14, 15, 17, 19	24	7

**Table 2 cancers-15-01793-t002:** Preoperative characteristics.

Studies	N	OPN	LPN	RAPN	Sex		Average age (Years) Total OPN LPN RAPN	BMI (Kg/m^2^) OPN LPN RAPN	ECOG ≥1 (%) Total OPN LPN RAPN	LATERALITY Predominant Total OPN LPN RAPN	DIAMETER Total OPN LPN RAPN	LOCATION Predominant OPN LPN RAPN	GROWTH PATTERN Predominant OPN LPN RAPN	Hb before OPN LPN RAPN	Cr before (mg/dL) OPN LPN RAPN	EGFR before (ml/min/m^2^) Total OPN LPN RAPN
					M (%) Total OPN LPN RAPN	F (%) Total OPN LPN RAPN										
Porpiglia et al., 2015 [10]	285	133	57	95	- 65.4 73.7 44.2	- 34.6 26.3 55.8	- 62.5 60 57.3	- 26 25.6 25.8	- 30.8 24.6 11.6	- Right Right Right	- 5 5 5	Superior Polar Mesorrenal Lower polar	Exophytic Exophytic Exofitico	14 14 14	0.9 1 1	- 87 86
Webb et al., 2015 [14]	66	21	31	14	- 66.67 59.38 42.86	- 33.3 46.62 57.14	- 53.6 55.53 605			- Left Left Same	- 4.22 2.7 2.9		Exophytic Exophytic Exophytic		0.95 0.9 0.8	
Kizilay et al., 2019 [15]	142	-	71	71	- - 52.2 56.4	- - 47.8 43.6	- - 54.6 52.9	- 23.8 24.5		- - Left Right	- - 2.79 2.48		- Exophytic Exophytic		- 0.88 0.92	- - 84.9 82.6
Zaid et al., 2017 [16]	1773	1407	196	170	63	37	61		8							
Ingels et al., 2021 [11]	191	69	17	105	61.3		78.4		41.3		4.7					72.7
Würnschimmel et al., 2020 [13]	115	-	54	61	- - 39 40	- - 15 21	- - 63.9 62.7	- 28.5 29.5		- Right Left	- 3.5 4.5					- - 78.8 80.4
Haseebuddin et al., 2010 [17]	38			38			- - - 62				- - - 3.3 (>2 cm) and 1.25 (<2 cm)					
Hinata et al., 2020 [12]	105			105	- - - 22		- - - 62.6	- - - 29.2			- - - 3.2	Hiliar	Endophytic		- - - 0.88	- - - 69

OPN: Open Partial Nephrectomy, LPN: Laparoscopic Partial Nephrectomy, RAPN: Robotic assisted Partial Nephrectomy, Hb: Hemoglobin, Cr: Creatinine.

**Table 3 cancers-15-01793-t003:** Intra and postoperative characteristics.

Studies	Total Operating Time (min) Total OPN LPN RAPN	Warm Ischemia Time (min) Total OPN LPN RAPN	Surgical Margins Positive (%) Total OPN LPN RAPN	Hospital Stay (days) Total OPN LPN RAPN	Blood Loss (cc) Total OPN LPN RAPN	Postoperative EGFR Difference Total OPN LPN RAPN	Death Total OPN LPN RAPN	Complications Postoperative (%) Total OPN LPN RAPN	Metastasis-Free Survival % Total OPN LPN RAPN	Recurrence-Free Survival (at 5 years) % Total OPN LPN RAPN
Porpiglia et al., 2015 [10]	- 135 129 155	- 16 24 18	- 6.8 1.9 2.5	-	- 200 200 150	- 8.7 7.3 1.6		- 12.8 1.8 2.1		
Webb et al., 2015 [14]	- - 158 210	- 30.69 24.07 28.01		- 4 2 3	- 300 100 150			- 0 4.5 0		
Kizilay et al., 2019 [15]	- - 158 176	- - 24.39 18.81	- - 3 2	- - 3.5 3.2	- - 240 210	- - 12.39 11.38			- - 68 69	- - 61 64
Zaid et al., 2017 [16]			51	4	200		0.1	6.7		
Ingels et al., 2021132021 [11]	150				200			17		
Würnschimmel et al., 2020 [13]	- - 192.3 230.2	- - 21.1 19.6	- - 0 5	- - 6.3 6.1		- - 14 16		- - 31 21		
Haseebuddin et al., 2010 [17]	- - - 131.9 (<2 cm) 145.8 (>2 cm)	- - - 21 (<2 cm) 24.7 (>2 cm)		2.5	- - - 130.9 (<2 cm) 136.4 (>2 cm)			13		
Hinata et al., 2020 [12]	- - - 146	- - - 20.2	- - - 1.9	- - -	- - - 138.6	- - - 0.19		- - - 63.8		

OPN: Open Partial Nephrectomy, LPN: Laparoscopic Partial Nephrectomy, RAPN: Robotic assisted Partial Nephrectomy.

## Data Availability

Not applicable.

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
