# Peer review of "Robotic versus Laparoscopic Partial Nephrectomy in the New Era: Systematic Review"

_cancers, 2023, doi:10.3390/cancers15061793_

Round 1

Reviewer 1 Report

INTRO:

- RPN adoption increased in the last decades, and indications have been expanded also to more complex and challenging cases; please cite PMID: 34973156 and PMID:27106494

- RPN favoured the reduction of WIT, with the widespread diffusion of minimally ischemic approach; please cite PMID: 31342589 and PMID: 30553632

DISCUSSION:

-You stated: "NIRF with ICG is considered a safe and practical tool that helps the surgeon to identify anatomical structures and also resection margins, enhancing renal function short-term preservation"; please cite PMID: 30262342

- You stated: "Trifecta is considered a new term to describe the surgical quality, that combines the following characteristics: the presence of negative surgical margins, the absence  of serious complications, and a ≤ 30% postoperative estimated glomerular filtration rate (eGFR) reduction; the correct reference is PMID: 31833720

Author Response

We added in our paper, these two statements:

- RPN adoption increased in the last decades, and indications have been expanded also to more complex and challenging cases – I added this reference PMID: 34973156 and PMID:27106494

- RPN favoured the reduction of WIT, with the widespread diffusion of minimally ischemic approach- I added these references PMID: 31342589 and PMID: 30553632

At discussion, we added these two references

-   "NIRF with ICG is considered a safe and practical tool that helps the surgeon to identify anatomical structures and also resection margins, enhancing renal function short-term preservation"- PMID: 30262342

- "Trifecta is considered a new term to describe the surgical quality, that combines the following characteristics: the presence of negative surgical margins, the absence  of serious complications, and a ≤ 30% postoperative estimated glomerular filtration rate (eGFR) reduction- We modified the reference for the correct one:  PMID: 31833720

Reviewer 2 Report

Dear Authors,

The manuscript "Robotic versus Laparoscopic Partial Nephrectomy in the new era: systematic review" by Guerrero has summarized the differences between RAPN and LPN. I have just a few suggestions.

1. Some background information and references are missing:

In introduction, page 1, Robotic approach is developed and used in many other fields, for example, breast cancer and reconstruction surgery. (please cite: 1. Efficacy of da Vinci robot-assisted lymph node surgery than conventional axillary lymph node dissection in breast cancer - A comparative study. Int J Med Robot. doi: 10.1002/rcs.2307.  2. Robot-Assisted Minimally Invasive Breast Surgery: Recent Evidence with Comparative Clinical Outcomes. J Clin Med. doi: 10.3390/jcm11071827.)

Best,

Author Response

We added some background information in introduction and its corresponding references:

Robotic approach is used in many other urological surgeries such as prostatectomy and it is widely developed in many other fields, for example, breast cancer and reconstruction surgery5,6

  1. Efficacy of da Vinci robot-assisted lymph node surgery than conventional axillary lymph node dissection in breast cancer - A comparative study. Int J Med Robot. doi: 10.1002/rcs.2307
  2. Robot-Assisted Minimally Invasive Breast Surgery: Recent Evidence with Comparative Clinical Outcomes. J Clin Med. doi: 10.3390/jcm11071827.)

Reviewer 3 Report

 This systematic review, which is well written, compared robot-assisted with laparoscopic partial nephrectomy. For smaller renal masses, partial nephrectomy is commonly accepted as superior to radical nephrectomy. The authors found a relatively low number of papers to meet the entrance criteria (<10/>1000). Only one of the papers is based on a randomized study, and the authors also describe a high variability between papers. The database therefore seems relatively weak to justify the robust conclusions, which support the robot-assisted approach, but more on the basis of its technical advantages and - at least for my feeling - less on the basis of the destilled results.

Author Response

We modified the conclusions and we remarked a high variability between papers included in our review.

“. However, only two of the papers are based on randomized studies, there are high variability between papers.”

“Robotics is an outstanding alternative to laparoscopy to perform a partial nephrectomy, which seem to provide great benefits for complex tumors treatment, leading to a WIT decline. RAPN also seem to improve oncologic control, surgeon ergonomics and also movement freedom, without increasing complications or worsening postoperative recovery. Finally, it is essential to the procedure planning according to the surgeon’s experience, patient characteristics and tumor morphology”.

Round 2

Reviewer 1 Report

No further comments

Reviewer 2 Report

Strongly suggest for publication.

Reviewer 3 Report

There is nothing to add to my previous comment.